# The Effect of Functional Biomechanics Garment for Walking

**DOI:** 10.3390/ijerph182312415

**Published:** 2021-11-25

**Authors:** Toshinori Miyashita, Sho Katayama, Ayane Yamamoto, Kodai Sakamoto, Masashi Kitano, Raita Takasaki, Shintarou Kudo

**Affiliations:** 1Inclusive Medical Science Research Institute, Morinomiya University of Medical Sciences, Nankokita 1–26–16, Suminoe Ward, Osaka 559-8611, Japan; miyashita.osaka@gmail.com (T.M.); 20154026sakamotokoudai@gmail.com (K.S.); 2Department of Rehabilitation, Meidaimae Orthopedic Clinic, 1–38–25, Matsubara Setagaya Ward, Tokyo 156-0043, Japan; s.katayama5218@gmail.com; 3Department of Rehabilitation, AR-Ex Oyamadai Orthopedic Clinic, Tokyo Arthroscopy Center, 4–13–1, Todoroki Setagaya Ward, Tokyo 158-0082, Japan; keitojunta4121@gmail.com; 4Graduate School of Health Sciences, Morinomiya University of Medical Sciences, Nankokita 1–26–16, Suminoe Ward, Osaka 559-8611, Japan; kitakita1215reha@yahoo.co.jp; 5Department of Acupuncture, Morinomiya University of Medical Sciences, Nankokita 1–26–16, Suminoe Ward, Osaka 559-8611, Japan; takasaki@morinomiya-u.ac.jp; 6AR-Ex Medical Research Center, 4–13–1, Todoroki Setagaya Ward, Tokyo 158-0082, Japan

**Keywords:** functional biomechanics garment, kinematics, kinetics, dynamic joint stiffness, biomechanical methods

## Abstract

The purpose of this study was to investigate the effects of a functional biomechanics garment (FBG) with a lower extremity assist function. 32 healthy male participants were included in this study. Participants were divided into an FBG with taping function group (FBG group) and a compression garment group (CG group). Cadence (steps/min), step length (m), and usual walking speed (m/s) were measured as spatio-temporal data. Kinetics, kinematics data, and dynamic joint stiffness (DJS) of the lower extremity were calculated using a three-dimensional gait analysis system. The FBG group showed significantly faster walking speed (FBG, 1.54 ± 0.12 m/s; CG, 1.42 ± 0.15 m/s, *p* < 0.05) and reduced hip DJS in terminal stance (FBG, 0.033 ± 0.014 Nm/kg/degree; CG: 0.049 ± 0.016 Nm/kg/degree, *p* < 0.05) compared to the CG group. The FBG decreased hip DJS in the terminal stance and affected walking speed. The passive elastic moment generated by the high elasticity part of the hip joint front in the FBG supported the internal hip flexion moment. Therefore, our FBG has a biomechanical effect. The FBG may be useful as a tool to promote health activities.

## 1. Introduction

Walking ability is important that affecting life-space mobility [1], promoting physical activity and activities of daily of living (ADL) [2,3], and improving or main-taining quality of life (QOL) [4,5]. Walking is often used as a means tool for health promotion and is a recommended physical activity [6]. In several studies, the physical benefits of walking have been described [7,8,9,10,11]. The step length and walking speed can be cited as indicator of the characteristics of the walking [12,13,14]. In addition, the step length and walking speed vary depending on the joint kinematics and joint kinetics of the lower extremity during walking [14,15,16]. Therefore, it is considered to increase the step length and maintain the walking speed in order to obtain the beneficial effect of walking, while reducing load on the joints of the lower extremity.

In recent years, a number of compression garments have been sold by many sports brands. The purpose of wearing a compression garment is to enhance performance in various sports, and the target audience has a wide range from sports beginners to professional athletes. Several studies have examined the influence or effect of compression garments [17,18,19]. These effects differ depending on the structure and material of the compression garments, including fatigue reduction [20,21], recovery enhancement [22,23], and proprioception facilitation [24]. Many studies have investigated compression garments for sports only, however, a few studies have examined biomechanical effects of compression garments used for walking.

In order to maintain walking speed, it is necessary to advance the center of gravity of the body forward and efficiently. For this reason, it is important that the ankle dorsiflexion muscles, hip and knee extension muscles, and hip abductor muscles activate in a timely manner during the stance phase [25,26]. Muscle activity of the ankle dorsiflexor muscles at initial contact accelerates the center of gravity upward, and the hip and knee extensor and hip abductor muscles on the ipsilateral side provide support in the early- to mid-single-leg stance [25,26]. Those muscle activities generated support moment which needed to achive efficient of the walking. Thus, in order to maintain a step length and a walking speed, it is necessary to activate muscles in a timely manner during the stance phase and at push-off during the late stance [26,27]. We have developed functional biomechanics garment (FBG) designed to support the kinematics and kinetics of the lower extremity based on the walking mechanism active during the stance phase.

The purpose of this study was to investigate the effects of functional biomechanics garment with lower extremity assist function using biomechanical assessment methods.

## 2. Materials and Methods

32 healthy male participants (age, 21.7 ± 3.1 years; height, 1.72 ± 0.54 m; weight, 64.1 ± 7.3 kg) were included in this study. Participants were healthy adult males with-out musculoskeletal or gait disorders. Exclusion criteria were participants with musculoskeletal or neuromuscular disorders or pain that affected their walking. When, during analysis, participant’s kinematic parameters fell outside of the normal ranges or could not be calculated due to noise of acceleration waveform, these data were excluded. This study was approved by the Morinomiya University of Medical Sciences ethics committee (Approval number: 2019−130), and all participants provided in-formed consent. Participants were divided into a functional biomechanics garment with taping function (FBG) group and a compression garment (CG) group.

All participants were required to perform 5 straight-line walking trials along a 10 m level walkway at their usual walking speed. A three-dimensional gait analysis system (Vicon MX system, Oxford Metrics Ltd., Oxford, England) was used for movement data acquisition and analysis based on standard procedures. The system consisted of eight infrared cameras (VICON VERO; 100 Hz) and two force plates (AMTI; 1000 Hz). Sixteen reflective markers were mounted on the skin according to the plug-in gait lower body model using double-sided adhesive tape. Cadence (steps/min), step length (m), and walking speed (m/s) were measured as spatio-temporal data. Kinetics and Kinematics data of the lower extremity were calculated using Nexus and filtered with a low-pass filter at 10 Hz. The peak values of the lower extremity (hip, knee, ankle) joint angles (degree), joint moments (Nm/kg), joint power (W), and Ground reaction force (Fz: Nm/kg) were calculated in the sagittal plane during gait cycle. Moreover, Dynamic Joint Stiffness (DJS) (Nm/kg/deg) at the lower extremity was calculated from joint angle and joint moment during walking in accordance with the method proposed by previous studies [28,29,30]. The phases of the gait cycle to calculate the DJS were Mid Stance and Pre Swing for the ankle joint, Terminal Stance and Initial Swing for the hip joint, and Loading Response for the knee joint.

### 2.1. The Compression Garments (CG)

There were used two garments in this study. Both the compression garments (CG) without taping function and the functional biomechanics garments (FBG) with the taping function are composed of 93% Nylon and 7% polyurethane with both garments have provide a pressure of 5.6–8.4 mmHg from the buttock to the heel. The CG and the FBG are made in the same color and shape so look alike that participants can’t tell which is which. In addition, both garments use stretchy yarn called “Woolly Nylon” which is made by twisted to nylon yarn. Points of the difference between the FBG and the CG are not material but the knitting and weaving methods.

The FBG is a compression garment that adheres from the buttock to the heel. The characteristics of the FBG are a mix of different elastic parts to support accurately lower extremity movement (i.e., joint moment) at the stance phase during walking. The FBG was designed in two important parts: elastic support and rigid support. The rigid and elastic rate of the FBG was changed by changing the knitting and weaving methods. The main support function of the lower extremity joint internal moment are as follows: (1) Ankle dorsal flexion moment support during loading response, (2) Hip ex-tension moment support during loading response, (3) Hip flexor moment support during Late stance, (4) Knee flexor moment support during Late stance and Terminal swing, (5) Ankle plantar flexion moment support during Late stance, and (6) Hip abduction moment support during loading response and Mid stance (Figure 1).

All participants wore shorts and comfortable walking was measured using VI-CON.

Participants were blinded to the function of tights and were classified into FBG group and control group. This was used as pre-intervention (i.e., baseline) data for the survey items. The size of the both types of tights is one-size-fits-all, and the investigator confirmed the size fit. All participants walked on the treadmill at a speed of 4.5 km/h for 5 min in order to get accustomed to the FBG tights walking support function before measuring comfortable walking. All participants used a treadmill to warm up under the same walking conditions such as walking speed and walking time. Immediately after treadmill walking finished, participants were measured during comfortable walking using VICON with the FBG or CG (Figure 2). The right lower extremity was analyzed.

### 2.2. Statistical Analyses

The characteristics of the participants and the pre-intervention data for each sur-vey item were used as a baseline, and an unpaired *t*-test between groups was performed. Next, in order to investigate the wearing effect of the pre-intervention and in the wearing each garments type (FBG group and the CG group), all parameters were compared between two conditions (pre and post) using paired *t*-test in each garments group. In addition, a repeated two-way ANOVA was performed to compare the garments effect between groups (FBG and CG) and intervention effect (between pre-garments and post-garments). When the main effect was observed, a *t*-test was performed to evaluate the differences between the two groups. All statistical analyses were performed using IBM SPSS Statistics for Windows (IBM SPSS Statistics for Windows, Version 24.0. Armonk, NY, USA). Values of *p* < 0.05 were considered to indicate statistical significance for all tests. The sample size required for this study was calculated using the power analysis software G * Power (ver. 3.1.9.4; Hein-rich-Heine-Universität Düsseldorf, Düsseldorf, Germany). A power analysis found that to detect a difference, a study population of 16 healthy participants per group would be required.

## 3. Results

There were no differences observed between the groups in any characteristic or performance variable at baseline (Table 1). In the spatio-temporal data, the increase in walking speed was the main effect of the garments. As a result of the *t*-test, the FBG group showed significantly faster walking speed than the CG group (*p* < 0.05). The decrease in the peak knee joint flexion angle during the initial-swing and the decrease in the peak knee extension moment during the loading-response showed a main effect of the intervention, however no significant difference in joint power was observed (Table 2, Table 3, Table 4 and Table 5). The decrease in hip DJS in the late stance was the main effect of the garments and the intervention. After a *t*-test, the FBG group showed significantly less hip DJS in late stance than the CG group (*p* < 0.05) (Table 6 and Table 7).

Two-way ANOVA showed a significant difference in the walking speed (Table 4) and hip DJS (Table 7) of the terminal stance between the FBG and CG groups (*p* < 0.05). There was no significant difference in the effect of interactions, respectively. No significant differences in the other variables were detected.

## 4. Discussion

The main objective of this study was to investigate the effects of FBG with lower extremity assist function. Wearing the FBG increased walking speed. In the kinematics and kinematics analysis, wearing the FBG increased the peak hip joint extension angle in the terminal stance, decreased the peak knee joint flexion angle in the initial swing, and decreased the peak hip joint flexion moment and hip joint DJS in the late stance. On the other hand, wearing the CG decreased the hip joint DJS in the late stance. Overall, the FBG increased walking speed and decreased hip joint DJS in the late stance more compared to the CG. Therefore, our results show that the FBG decreased hip joint DJS in the late stance and affected walking speed.

A large number of studies of the effects of compression garments have investigated professional players and amateur athletes [19,31,32,33]. In recent systematic reviews, several studies targeted runners and athletes and there are many reports on physiological effects, physical effects, and impact on running performance. Other studies have suggested that compression garments help to maintain muscle temperature, reduce exhaustion, and reduce edema and swelling, as well as to have psychological effects [17,18]. Little has been reported on the mechanism of action of compression garments using biomechanical methods. Chen et al. reported that compression stockings increase ankle dorsiflexion moment and peak knee extension moment during the early stance, but there was no significant difference observed in spatiotemporal data such as walking speed [34]. Although the CG used in this study also confirmed the effect of reducing hip joint DJS in the late stance, it did not show significant difference the spatiotemporal data. Meanwhile, the FBG was able to decrease the hip joint DJS in the late stance and increase the walking speed. Therefore, it became clear that our FBG has a biomechanical effect.

Dynamic joint stiffness (DJS) is defined as the resistance that a joint offers during walking in response to an applied moment [28]. The DJS during walking calculated as an angular coefficient of linear regression of the plot of the hip flexion moment during a given stance phase of walking versus hip extension angle in the same stance phase [28,29]. According to previous research, the hip DJS is expressed by plotting the values of hip flexion moment versus hip extension angle during late stance, from onset of the hip flexion moment to the angle at which the hip reaches its peak extension [29]. In the biomechanics of normal walking, passive stretch of the hip joint front as the hip extension angle increases in the late stance. Moreover, the ground reaction force vector passes through the hip joint and an internal hip flexion moment occurs in the late stance [25]. In this study, the FBG significantly decreased the internal hip flexion moment in the late stance and increased the peak hip extension angle by 2.7 degrees. The hip joint front part in the FBG was designed with a highly elastic part. Thus, the support function of the FBG was able to increase the hip extension angle while decreasing the internal hip flexion moment. Consequently, by these biomechanical changes may have affected walking speed. The reason for the decrease in DJS in the late stance may be due to the hip joint part structure of the FBG, which is compression with a taping function based on the mechanism of walking.

The present study demonstrated that the FBG may be able to maintain step length and prevent a decline in walking speed in healthy persons. Preventing a decline in walking speed can be expected to reduce the risk of adverse events and enjoyment the beneficial effects of walking. Moreover, for patients or subjects who require continuous coach and exercise therapy, it may be possible to practice correct walking training just by wearing the FBG. The FBG may be useful as a tool to easily promote in health promotion activities.

There are several limitations of this study. First, the participants of this study were healthy adults and only men. Additionally, it might be difficult to obtain changes to examine the kinematic and kinetic effects of low-intensity activity such as walking. Therefore, further research on middle-aged to elderly people would strengthen the effect of the FBG. Next, this study was examined for immediate effects, however, the long-term effects are unclear because it was not a longitudinal study. Finally, the effect of compression garments on movement and the mechanism of action are unclear, such as the material of compression tights, elasticity [35], and proprioceptors [36,37]. In future studies, electromyography should be used to clarify the mechanism of effect.

## 5. Conclusions

Wearing the FBG increase walking speed, and the decrease in the peak knee joint flexion angle during the initial-swing and the decrease in the peak knee extension moment during the loading-response. In addition, the FBG decreased hip DJS in the terminal stance. The passive elastic moment by the high elasticity part of the hip joint front in the FBG supported the internal hip flexion moment. For the above reasons, our FBG has a biomechanical effect. Therefore, the FBG may be useful as a tool to support the walking.

## Figures and Tables

**Figure 1 ijerph-18-12415-f001:**
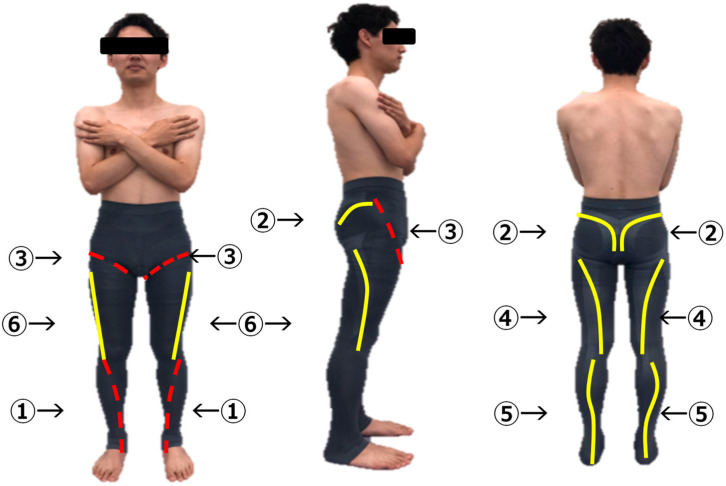
The FBG is designed for two important parts which are elastic support part and rigid support part. The elastic support part had designed by knitting and weaving with high elastic (solid line). The rigid part had designed by knitting and weaving with low elastic (dashed line). The main support function of lower extremity joint internal moment are as follows: ➀ Ankle dorsal flexion moment support during loading response, ➁ Hip extension moment support during loading response, ➂ Hip flexor moment support during Late stance, ➃ Knee flexor moment support during Late stance and Terminal swing, ➄ Ankle plantar flexion moment support during Late stance, ➅ Hip abduction moment support during loading response and Mid stance.

**Figure 2 ijerph-18-12415-f002:**
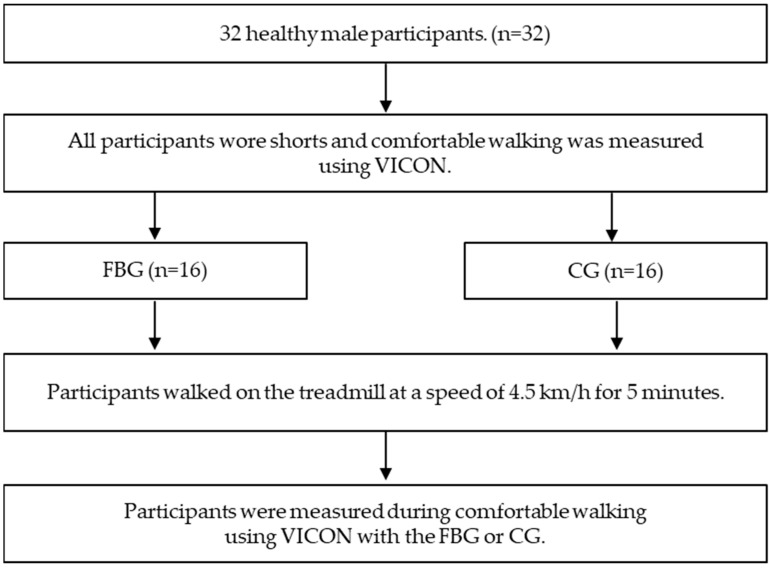
Experimental procedure.

**Table 1 ijerph-18-12415-t001:** Comparing baseline characteristics between groups. average (SD), *p* values < 0.05 are bolded.

	FBG	CG	*p* Value
Age (year)	22.6 (3.7)	22.0 (3.7)	0.67
Height (cm)	172.1 (6.1)	171.0 (5.6)	0.26
Body Weight (kg)	62.8 (5.8)	63.9 (8.2)	0.61
BMI	21.2 (1.6)	22.2 (2.4)	0.17

**Table 2 ijerph-18-12415-t002:** Spatio-temporal data during walking pre and post wearing of FBG. average (SD), paired *t*-test, *p* values < 0.05 are bolded.

	Pre	FBG	*p* Value
**Spatio-temporal data**			
Cadence (steps/min)	**121.3 (7.0)**	**126.7 (7.6)**	**<0.01**
Step Length (m)	**0.69 (0.05)**	**0.73 (0.06)**	**<0.01**
Walking Speed (m/s)	**1.44 (0.12)**	**1.54 (0.12)**	**<0.01**

**Table 3 ijerph-18-12415-t003:** Kinematics Kinetics data, and Ground reaction force during walking pre and post wearing of FBG. average (SD), paired *t*-test, *p* values < 0.05 are bolded.

	Pre	FBG	*p* Value
**ROM (°)**			
Peak ankle plantar flexion (toe-off)	−16.3 (5.5)	−16.7 (4.3)	0.69
Peak hip extension (late stance)	**−16.8 (5.3)**	**−19.5 (5.2)**	**<0.01**
Peak knee flexion (initial swing)	**63.2 (5.3)**	**56.1 (8.4)**	**<0.05**
**Joint Moment (N/m/kg)**			
Ankle plantar flexion (push off)	0.023 (0.004)	0.022 (0.003)	0.48
Hip extension (late stance and early swing)	**−0.017 (0.002)**	**−0.015 (0.004)**	**<0.05**
Knee extension (loading response)	0.011 (0.005)	0.008 (0.005)	0.07
**Joint Power (W)**			
Ankle: A (push off)	4.22 (0.49)	4.10 (0.67)	0.44
Knee 1: K1 (loading response)	−1.39 (0.76)	−1.22 (0.71)	0.48
Knee 2: K2 (late stance and early swing)	−1.27 (0.47)	−1.30 (0.36)	0.86
Hip: H (late stance and early swing)	1.42(0.29)	1.50(0.43)	0.35
**Ground Reaction Force (Nm/kg)**			
Fz: First Peak	0.179 (0.030)	0.184 (0.028)	0.07
Fz: Second Peak	0.175 (0.023)	0.174 (0.023)	0.20

**Table 4 ijerph-18-12415-t004:** Spatio-temporal data, during walking under two garments condition, i.e., effect of wearing comparison FBG and CG, average (SD), two-way ANOVA, *p* values < 0.05 are bolded.

	FBG		CG		Main Effect		Interaction
	Pre	Post	Pre	Post	Garments	Pre-Post	
**Spatio-temporal data**							
Cadence (steps/min)	121.3 (7.0)	126.7 (7.6)	122.0 (5.1)	122.9 (5.6)	0.343	0.053	0.171
Step Length (m)	0.69 (0.05)	0.73 (0.06)	0.70 (0.06)	0.71 (0.07)	0.869	0.136	0.422
Walking Speed (m/s)	**1.44 (0.12)**	**1.54 (0.12)**	**1.40 (0.13)**	**1.42 (0.15)**	**0.024**	0.065	0.204

**Table 5 ijerph-18-12415-t005:** Kinematics, Kinetics data and Ground reaction force during walking under two garments condition, i.e., effect of wearing comparison FBG and CG, average (SD), two-way ANOVA, *p* values < 0.05 are bolded.

	FBG		CG		Main Effect		Interaction
	Pre	Post	Pre	Post	Garments	Pre-Post	
**ROM (°)**							
Peak ankle plantar flexion (toe-off)	−16.3 (5.5)	−16.7 (4.3)	−16.2 (6.8)	−17.9 (7.1)	0.705	0.492	0.662
Peak hip extension (late stance)	−16.8 (5.3)	−19.5 (5.2)	−17.6 (7.4)	−19.5 (6.8)	0.781	0.143	0.774
Peak knee flexion (initial swing)	**63.2 (5.3)**	**56.1 (8.4)**	**59.7 (4.5)**	**55.7 (9.8)**	0.291	**0.003**	0.394
**Joint Moment (N/m/kg)**							
Ankle plantar flexion (push off)	0.023 (0.004)	0.022 (0.003)	0.023 (0.004)	0.023 (0.004)	0.685	0.529	1.000
Hip extension (late stance and early swing)	−0.017 (0.002)	−0.015 (0.004)	−0.015 (0.003)	−0.015 (0.005)	0.332	0.140	0.167
Knee extension (loading response)	**0.011** **(0.005)**	**0.008** **(0.005)**	**0.009** **(0.005)**	**0.008** **(0.005)**	0.291	**0.003**	0.394
**Joint Power (W)**							
Ankle: A (push off)	4.22 (0.49)	4.10 (0.67)	4.13 (1.00)	4.22 (1.16)	0.933	0.949	0.628
Knee 1: K1 (loading response)	−1.39 (0.76)	−1.22 (0.71)	−1.24 (0.81)	−1.00 (0.75)	0.332	0.285	0.833
Knee 2: K2 (late stance and early swing)	−1.27 (0.47)	−1.30 (0.36)	−1.06 (0.27)	−1.17 (0.32)	0.063	0.442	0.646
Hip: H (late stance and early swing)	1.42 (0.29)	1.50 (0.43)	1.31 (0.27)	1.47 (0.37)	0.422	0.176	0.628
**Ground Reaction Force (Nm/kg)**							
Fz: First Peak	0.179 (0.030)	0.184 (0.028)	0.180 (0.019)	0.184 (0.019)	0.946	0.505	0.894
Fz: Second Peak	0.184 (0.028)	0.184 (0.028)	0.167 (0.024)	0.171 (0.023)	0.355	0.850	0.627

**Table 6 ijerph-18-12415-t006:** Dynamic joint stiffness for pre and post wearing of FBG. Post wearing of the FBG had significantly decrease dynamic joint stiffness values in the Terminal stance when compared to the pre wearing condition. average (SD), paired *t*-test (*p* < 0.05). *p* values < 0.05 are bolded.

		Pre	FBG	*p* Value
**Dynamic Joint Stiffness (Nm/kg/deg)**				
**Ankle**	Mid Stance	0.053 (0.022)	0.049 (0.022)	0.13
	Pre Swing	0.053 (0.011)	0.054 (0.013)	0.92
**Hip**	Terminal Stance	**0.051 (0.024)**	**0.033 (0.014)**	**<0.01**
	Initial Swing	0.019 (0.006)	0.019 (0.005)	0.70
**Knee**	Loading Response	0.075 (0.036)	0.073 (0.043)	0.76

**Table 7 ijerph-18-12415-t007:** Dynamic joint stiffness during walking under two garments condition, i.e., effect of wearing comparison FBG and CG. The FBG group had significantly lower Hip Dynamic joint stiffness values in the Terminal stance than the CG groups. average (SD), two-way ANOVA, *p* values < 0.05 are bolded.

		FBG		CG		Main Effect		Inter Action
		Pre	Post	Pre	Post	Garments	Pre-Post	
**Dynamic Joint Stiffness (Nm/kg/deg)**								
**Ankle**	Mid Stance	0.053 (0.022)	0.049 (0.022)	0.051 (0.016)	0.048 (0.013)	0.733	0.517	0.926
	Pre Swing	0.053 (0.011)	0.054 (0.013)	0.057 (0.010)	0.055 (0.010)	0.346	0.751	0.652
**Hip**	Terminal Stance	**0.051** **(0.024)**	**0.033** **(0.014)**	**0.055** **(0.015)**	**0.049** **(0.016)**	**0.026**	**0.009**	0.185
	Initial Swing	0.019 (0.006)	0.019 (0.005)	0.019 (0.004)	0.020 (0.005)	0.712	1.000	0.687
**Knee**	Loading Response	0.075 (0.036)	0.073 (0.043)	0.064 (0.025)	0.064 (0.025)	0.236	0.913	0.876

## Data Availability

The data presented in this study are available on request from the corresponding author.

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
