# Peer review of "The Effect of Functional Biomechanics Garment for Walking"

_ijerph, 2021, doi:10.3390/ijerph182312415_

Round 1
Reviewer 1 Report
The presented article contains a very interesting topic, the solution of which is very meritorious. Several studies have already dealt with the topic, as the authors correctly state, but there are still a number of unexplored topics in the given area.
The work deserves publication, but it is necessary to make several adjustments and additions to make the text more comprehensible.
I recommend making the following additions and adjustments:
- the introduction would be appropriate only to relate to the topic addressed in the work, too general and broad statements - eg l. 32-35 are unnecessary; it would be more appropriate to remove this passage and pay more attention to the issue of walking
- in many places in the text the statements are too clear without documented sources, citations; I recommend revising the whole text and supporting citations (eg l. 50-52; l 69-72)
- line 53 is a sentence about the effects on the length and speed of walking, in addition to the above factors, there are many more, it would be appropriate to add a citation, source
- Line 172 lists patients, but studies are for healthy individuals only
- after reading the methodological procedures, I have a number of ambiguities, the methodological procedures should be better described; for example, it is not entirely clear why walking was on the treadmill and then outside it, if we are dealing with walking speed and the treadmill is set, proto it would therefore be appropriate to express the whole procedure better and concisely so as not to misunderstand the whole methodology
- data processing is satisfactory
- the discussion is sufficiently extensive, perhaps too broad, it would be better to focus specifically on their own findings and their justification, the authors are too concerned with considering what the material for the lower limbs that were applied in the study could be used
- conclusion - this passage needs to be fundamentally reworked; what the authors have arrived at must be clear here, the last sentence I quote "FBG can be useful as a tool to support health activities." - is again a consideration and not the result that the authors have arrived at
- I also recommend language proofreading
Author Response
Dear. Reviewer 1,
Thank you very much for reviewing our manuscript and offering valuable advice. Thank you in advance for your cooperation.
Please see the attachment.
Kind regards,
Toshinori Miyashita
Response to Reviewer 1 Comments
Point 1: the introduction would be appropriate only to relate to the topic addressed in the work, too general and broad statements - eg l. 32-35 are unnecessary; it would be more appropriate to remove this passage and pay more attention to the issue of walking
Response 1: I agree with your comment. I deleted and revised the sentence.
I changed sentence of the introduction, and significantly revisions in introduction.
Deleted Lines 32-36: Physical inactivity (insufficient physical activity) has been reported as a leading risk factor for death worldwide. Health status can decrease and many chronic diseases can result if physical inactivity continues [1]. As a result, health expenditures increase, and, with time, social problems develop. Therefore, strategies to increase physical activity and health promotion are being investigated.
Revised Lines35-37: Walking ability is important that affecting life-space mobility [1], promoting physical activity and activities of daily of living (ADL) [2, 3], and improving or maintaining quality of life (QOL) [4, 5].
I revised a redundant sentence of introduction.
Deleted Lines
38-50: In addition, walking does not require special implementation and can be easy to per-form. For these reasons, walking is easy for physicians, physical therapists, or health professions to promote.
On the other hand, previous studies have suggested a relationship between walking and degenerative joint complaints of the lower extremity. For example, the daily cumulative load for the knee joint has been suggested to be associated with knee osteoarthritis [7]. In a study of hip osteoarthritis, the daily cumulative load of the hip joint in the sagittal plane was reported to be associated with X-ray findings of secondary hip osteoarthritis [8]. The characteristics of the daily cumulative load for lower extremity joints depends on the number of steps per day. Therefore, if walking is to continue, in-creasing stride and reducing the number of steps per day is important for reducing the cumulative load. In addition, a decline in walking speed has been reported to be associated with many adverse events [9-12].
53-62: Step length and walking speed vary depending on the joint kinematics and joint kinetics of the lower extremity during walking. In particular, in order to improve step length and walking speed, several studies have examined the late stance phase during walking. In general, the ankle plantar flexor moment increases in the late stance phase and push-off is generated by ankle plantar flexor muscle activity. By generating push-off, the lower extremity swings forward and the step length can be obtained [13-15]. In addition, when generating the push-off, it is important to have an angle consisting of a line connecting the greater trochanter and the fifth metatarsal head at toe-off intersecting the laboratory’s vertical axis at late stance in the sagittal plane, namely, the Trailing Limb Angle [16]. Furthermore,
68-69: Hip abductors help overcome the downwardly accelerating center of gravity during mid-stance phase [17-19].
71-72: These walking mechanisms are considered the basics of walking.
81-82: We considered that walking compression garments may be used in walking training on a daily basis.
Added Lines 39-42:
The step length and walking speed can be cited as indicator of the characteristics of the walking [14-16]. In addition, the step length and walking speed vary depending on the joint kinematics and joint kinetics of the lower extremity during walking [16-18].
Revise and moved the line 45-148:
In recent years, a number of compression garments have been sold by many sports brands. The purpose of wearing a compression garment is to enhance performance in various sports, and the target audience has a wide range from sports beginners to professional athletes. Several studies have examined the influence or effect of compression garments [17-19]. These effects differ depending on the structure and material of the compression garments, including fatigue reduction [20, 21], recovery enhancement [22, 23], and proprioception facilitation [24]. Many studies have investigated compression garments for sports only, however, a few studies have examined biomechanical effects of compression garments used for walking.
Point 2: in many places in the text the statements are too clear without documented sources, citations; I recommend revising the whole text and supporting citations (eg l. 50-52; l 69-72)
Response 2: I revised sentence and added the reference.
Lines44-46: I revised sentence.
Therefore, it is necessary considered to increase the step length and maintain the walking speed in order to obtain the beneficial effect of walking, while reducing load on the joints of the lower extremity.
I added the reference.
Lines 156-158: in order to maintain a step length and a walking speed, it is necessary to activate muscles in a timely manner during the stance phase and at push-off during the late stance [26, 27]
(Added reference)
- Anderson, F.C.; Pandy, M.G. Individual Muscle Contributions to Support in Normal Walking. Gait Posture 2003, 17, 159–169.
- Liu, M.Q.; Anderson, F.C.; Schwartz, M.H.; Delp, S.L. Muscle Contributions to Support and Progression over a Range of Walking Speeds. J. Biomech. 2008, 41, 3243–3252, doi:10.1016/j.jbiomech.2008.07.031.
Point 3: line 53 is a sentence about the effects on the length and speed of walking, in addition to the above factors, there are many more, it would be appropriate to add a citation, source
Response 3: I added the reference.
Lines 39-42; the step length and walking speed vary depending on the joint kinematics and joint kinetics of the lower extremity during walking [14-16].
(Added reference)
- Fukuchi, C.A.; Fukuchi, R.K.; Duarte, M. Effects of Walking Speed on Gait Biomechanics in Healthy Participants: A Systematic Review and Meta-Analysis. Rev. 2019, 8, 1–11, doi:10.1186/s13643-019-1063-z.
- Judge, J.O.; Davis, R.B.; Ounpuu, S. Step Length Reductions in Advanced Age The Role of Ankle and Hip Kinetics. 1996, 51, 303–312.
- Ko, S.; Stenholm, S.; Metter, E.J.; Ferrucci, L. Age-Associated Gait Patterns and the Role of Lower Extremity Strength - Results from the Baltimore Longitudinal Study of Aging. Gerontol. Geriatr. 2012, 55, 474–479, doi:10.1016/j.archger.2012.04.004.
Point 4: Line 172 lists patients, but studies are for healthy individuals only
Response 4: I revised a part of the sentence.
Lines 331: a study population of 16 patients healthy participants per group would be required.
Point 5: - after reading the methodological procedures, I have a number of ambiguities, the methodological procedures should be better described; for example, it is not entirely clear why walking was on the treadmill and then outside it, if we are dealing with walking speed and the treadmill is set, proto it would therefore be appropriate to express the whole procedure better and concisely so as not to misunderstand the whole methodology
Response 5: I added the following sentence.
Revised Lines 312-315: All participants walked on the treadmill at a speed of 4.5 km/h for 5 minutes in order to get accustomed to the FBG tights walking support function before measuring comfortable walking. All participants used a treadmill to warm up under the same walking conditions such as walking speed and walking time.
Point 6: the discussion is sufficiently extensive, perhaps too broad, it would be better to focus specifically on their own findings and their justification, the authors are too concerned with considering what the material for the lower limbs that were applied in the study could be used
Response 6: I changed sentence of the discussion, and significantly revisions in discussion.
Revised lines: 412-425
Point 7: conclusion - this passage needs to be fundamentally reworked; what the authors have arrived at must be clear here, the last sentence I quote "FBG can be useful as a tool to support health activities." - is again a consideration and not the result that the authors have arrived at
Response 7: Revised lines 494: The FBG may be useful as a tool to promote health activities support the walking.
Point 8: - I also recommend language proofreading
Response 8: Thank you for your comment.
This manuscript has been reviewed by English proofreading.

Reviewer 2 Report
Dear Authors,
The quality of the presentation and the clarity of the text must be improved.
Please follow the comments below.
ABSTRACT:
Lines 25-26) "The passive elastic moment by the high elasticity part of the hip joint front in the FBG supported the internal hip flexion moment." -> "The passive elastic moment GENERATED by the high elasticity part of the hip ..."
In ABSTRACT, MATERIAL AND METHODS and FIGURE 2: Remove "thirty-two limbs of 32 healthy male participants" and replace with "32 healthy male participants".
INTRODUCTION:
Lines 46) "The characteristics of the daily cumulative load for lower extremity joints depends on..." Replace with "Overall, the daily cumulative load for lower extremity joints depends on..."
MATERIAL AND METHODS:
Lines 103-104) "Sixteen reflective markers were mounted on the skin according to the Plug in Lower Body Model" -> "plug-in gait lower body models".
Lines 112-115) "The phases of the lower extremity joint and the gait cycle to calculate the DJS were Mid Stance and Pre Swing of the ankle joint, Terminal Stance and Initial Swing of the hip joint, and Loading Response of the knee joint." Replace with: "The phases of the gait cycle to calculate the DJS were Mid Stance and Pre Swing FOR the ankle joint, Terminal Stance and Initial Swing FOR the hip joint, and Loading Response FOR the knee joint."
RESULTS:
I think you repeated some concepts twice: line 186-188 "Unpaired t-test showed significant differences between the FBG group and the CG group (p<0.05) in the walking speed and hip DJS of the terminal stance."
You have already said that in lines 176-177 "As a result of the t-test, the FBG group showed significantly faster walking speed than the CG group (p<0.05)." and in lines 182-183: "After a t-test, the FBG group showed significantly less hip DJS in late stance than the CG group (p<0.05)."
Line 184) "Two-way analysis of variance" better: "Two-way ANOVA".
Table 2 and Table 4 captions: "Table 2 (4). Pre and post wearing of FBG, average (SD), un-paired t-test (p< 0.05)." the is PAIRED t-test (statical analysis to test two related observations (i.e., two observations per subject)).
Table 5 caption "Table 5. Effect of wearing comparison FBG and CG DJS average (SD), ANOVA (p<0.05)." Is two-way ANOVA.
Moreover, for all the tables I suggest writing the statistically significant lines in bold.
DISCUSSION:
In general, the discussion section is not very clear. Try to rephrase it extensively. Try to not repeat the same concepts more than once. Keep the sentences shorter and simpler.
Lines 200-202) "On the other hand, wearing the CG decreased the hip joint DJS in the late stance. Furthermore, the FBG increased walking speed and decreased in hip joint DJS in the late stance compared to the GC. " All these sentences are not very clear. What do you mean here?
That the decrease of the hip joint DJS with the FBG is greater than the decrease of the hip joint DJS with the CG? Try like that: "On the other hand, wearing the CG decreased the hip joint DJS in the late stance. Overall, the FBG increased walking speed and decreased hip joint DJS in the late stance more compared to the GC. "
Line 212) "On the other hand, Chen et al. reported that compression ..."
Remove on the other hand. "Chen et al. reported that compression"
Lines 220-222) "The DJS is a measure of joint stiffness during gait divided by the change in joint moment in a given stance phase of gait by the change in joint angle in the same stance phase [28, 29]." This sentence is not clear and the definition of the DJS is wrong. Replace with: "The DJS during gait is calculated as an angular coefficient of linear regression of the plot of the hip flexion moment during a given stance phase of gait versus hip extension angle in the same stance phase [28, 29]."
Lines 228-234) how are these sentences "In addition, passive stretch of the hip joint front as the hip extension angle increases, increases the ratio of passive elastic moments to the percentage of hip flexion moment [35]. It has been shown that stretching of the hip flexors and tendon occurs since passive tissues are affected by the swing of the lower extremity during early swing phase [36]. Therefore, these biomechanical changes may affect walking speed" related to this: "The reason for the decrease in DJS in the late stance may be due to the hip joint part structure of the FBG, which is compression with a taping function based on the mechanism of walking."
Lines 235-237) "An increase peak hip extension angle in the late stance causes the hip joint front to stretch and an increase OF the passive elastic moment..."
Lines 244-246) "However, the FBG has structure that supports the activity of hip extensors, knee extensors, and ankle dorsiflexors, which are antagonistic to these muscles and the activity of hip extensors, knee extensors, and ankle dorsiflexors during the early stance." This sentence has no sense.
Lines 252-253) "In addition, preventing a decline in walking speed can be expected to reduce the risk of adverse events and enjoyment the beneficial effects of walking." Remove in addition: "Preventing a decline in walking speed can be expected to reduce the risk of adverse events and ENJOY the beneficial effects of walking."
CONCLUSIONS:
I think that here you can extend a bit more the sentence "The FBG may be useful as a tool to promote health activities." and thus emphasize the relevance of application of the FBG.
Moreover, in this section you can elaborate on the importance of this work or suggest extensions.
Bibliography:
3. author name: uppercase letter. remove the first name to be consistent with the other articles.
6. article title is in uppercase
Author Response
Dear. Reviewer 2,
We are grateful for the time and energy you expended on our behalf and for providing these insights.
Kind regards,
Toshinori Miyashita
Response to Reviewer 2 Comments
ABSTRACT
Point 1: Lines 25-26) "The passive elastic moment by the high elasticity part of the hip joint front in the FBG supported the internal hip flexion moment." -> "The passive elastic moment GENERATED by the high elasticity part of the hip ..."
Response 1: I revised a part of the sentence.
Lines 28-29:
The passive elastic moment generated by the high elasticity part of the hip joint front in the FBG supported the internal hip flexion moment.
In ABSTRACT, MATERIAL AND METHODS and FIGURE 2
Point 2: -: Remove "thirty-two limbs of 32 healthy male participants" and replace with "32 healthy male participants".
Response 2: I removed a part of the sentence.
Lines 20: Thirty-two limbs of 32 healthy male participants were included in this study.
Lines 164: Thirty-two limbs of 32 healthy male participants
Figure 2: Thirty-two limbs of 32 healthy male participants
MATERIAL AND METHODS:
Point 3: - Lines 103-104) "Sixteen reflective markers were mounted on the skin according to the Plug in Lower Body Model" -> "plug-in gait lower body models".
Response 3: I revised a part of the sentence.
Lines 179:
Sixteen reflective markers were mounted on the skin according to the Pplug-in Llower Bbody Mmodels using double-sided adhesive tape.
Point 4: - Lines 112-115) "The phases of the lower extremity joint and the gait cycle to calculate the DJS were Mid Stance and Pre Swing of the ankle joint, Terminal Stance and Initial Swing of the hip joint, and Loading Response of the knee joint." Replace with: "The phases of the gait cycle to calculate the DJS were Mid Stance and Pre Swing FOR the ankle joint, Terminal Stance and Initial Swing FOR the hip joint, and Loading Response FOR the knee joint."
Response 4: I revised a part of the sentence.
Lines 187-190:
The phases of the lower extremity joint and the gait cycle to calculate the DJS were Mid Stance and Pre Swing of for the ankle joint, Terminal Stance and Initial Swing of for the hip joint, and Loading Response of for the knee joint.
RESULTS:
Point 5: - I think you repeated some concepts twice: line 186-188 "Unpaired t-test showed significant differences between the FBG group and the CG group (p<0.05) in the walking speed and hip DJS of the terminal stance."
You have already said that in lines 176-177 "As a result of the t-test, the FBG group showed significantly faster walking speed than the CG group (p<0.05)." and in lines 182-183: "After a t-test, the FBG group showed significantly less hip DJS in late stance than the CG group (p<0.05)."
Response 5: I deleted the sentence.
Lines 360-361:
Un-paired t-test showed significant differences between the FBG group and the CG group (p<0.05) in the walking speed and hip DJS of the terminal stance.
Point 6: - Line 184) "Two-way analysis of variance" better: "Two-way ANOVA".
Response 6: I revised a part of the sentence.
Lines 358:
Two-way analysis of variance ANOVA
Point 7: - Table 2 and Table 4 captions: "Table 2 (4). Pre and post wearing of FBG, average (SD), un-paired t-test (p< 0.05)." the is PAIRED t-test (statical analysis to test two related observations (i.e., two observations per subject)).
Table 5 caption "Table 5. Effect of wearing comparison FBG and CG DJS average (SD), ANOVA (p<0.05)." Is two-way ANOVA.
Response 7: Thank you for your comment. I revised the error.
Table 2. Pre and post wearing of FBG, average (SD), un- paired t-test (p< 0.05).
Table 4. Pre and post wearing of FBG:DJS, average (SD), un- paired t-test (p< 0.05).
Table 5. Effect of wearing comparison FBG and CG DJS average (SD), two-way ANOVA (p<0.05)
Point 8: - Moreover, for all the tables I suggest writing the statistically significant lines in bold.
Response 8: I revised tables and added in bold.
DISCUSSION: In general, the discussion section is not very clear. Try to rephrase it extensively. Try to not repeat the same concepts more than once. Keep the sentences shorter and simpler.
Point 9: - Lines 200-202) "On the other hand, wearing the CG decreased the hip joint DJS in the late stance. Furthermore, the FBG increased walking speed and decreased in hip joint DJS in the late stance compared to the GC. " All these sentences are not very clear. What do you mean here?
That the decrease of the hip joint DJS with the FBG is greater than the decrease of the hip joint DJS with the CG? Try like that: "On the other hand, wearing the CG decreased the hip joint DJS in the late stance. Overall, the FBG increased walking speed and decreased hip joint DJS in the late stance more compared to the GC. "
Response 9: I agree with your comment. I have revised sentence as your suggested.
Lines 392-395:
On the other hand, wearing the CG decreased the hip joint DJS in the late stance. Furthermore Overall, the FBG increased walking speed and decreased in hip joint DJS in the late stance more compared to the CG.
Point 10: - Line 212) "On the other hand, Chen et al. reported that compression ..."
Remove on the other hand. "Chen et al. reported that compression"
Response 10: I deleted the sentence.
Lines 404:
On the other hand, Chen et al. reported that compression stockings increase….
Point 11: - Lines 220-222) "The DJS is a measure of joint stiffness during gait divided by the change in joint moment in a given stance phase of gait by the change in joint angle in the same stance phase [28, 29]." This sentence is not clear and the definition of the DJS is wrong. Replace with: "The DJS during gait is calculated as an angular coefficient of linear regression of the plot of the hip flexion moment during a given stance phase of gait versus hip extension angle in the same stance phase [28, 29]."
Response 11: I revised the sentence.
Lines 412-414:
The DJS during gait calculated as an angular coefficient of linear regression of the plot of the hip flexion is a measure of joint stiffness during gait divided by the change in joint moment in during a given stance phase of gait versus hip extension by the change in joint angle in the same stance phase [30, 31].
Point 12: Lines 228-234) how are these sentences "In addition, passive stretch of the hip joint front as the hip extension angle increases, increases the ratio of passive elastic moments to the percentage of hip flexion moment [35]. It has been shown that stretching of the hip flexors and tendon occurs since passive tissues are affected by the swing of the lower extremity during early swing phase [36]. Therefore, these biomechanical changes may affect walking speed" related to this: "The reason for the decrease in DJS in the late stance may be due to the hip joint part structure of the FBG, which is compression with a taping function based on the mechanism of walking."
Response 12: I removed redundant sentence and revised.
Deleted Lines
228-232: In addition, passive stretch of the hip joint front as the hip extension angle increases, increases the ratio of passive elastic moments to the percentage of hip flexion moment [35]. It has been shown that stretching of the hip flexors and tendon occurs since pas-sive tissues are affected by the swing of the lower extremity during early swing phase [36]. Therefore
235-249: An increase peak hip extension angle in the late stance causes the hip joint front to stretch and an increase the passive elastic moment of the FBG high elasticity part. In the late stance, the ground reaction force vector passes through the hip joint and an internal hip flexion moment occurs. The passive elastic moment of the high elasticity part of the hip joint front part in the FBG supports the internal hip flexion moment. This is suggested to decrease the hip DJS. On the other hand, an increase in walking speed is assumed to affect the kinetics in the early stance. As the walking speed in-creases, the ground reaction force vector is increased from initial contact to loading response, and increased ground reaction forces may increase the hip flexion moment, knee flexion moment, and ankle plantar flexion moment. However, the FBG has struc-ture that supports the activity of hip extensors, knee extensors, and ankle dorsiflexors, which are antagonistic to these muscles and the activity of hip extensors, knee exten-sors, and ankle dorsiflexors during the early stance. Therefore, these kinetic data did not significantly differ in the early stance. In other words, the FBG decreased hip DJS and increased walking speed by the support of the elasticity part at the hip joint front.
Added sentence Lines 417-420:
In the biomechanics of normal walking, passive stretch of the hip joint front as the hip extension angle increases in the late stance. Moreover, the ground reaction force vector passes through the hip joint and an internal hip flexion moment occurs in the late stance [27].
Point 13: - Lines 235-237) "An increase peak hip extension angle in the late stance causes the hip joint front to stretch and an increase OF the passive elastic moment..."
Response 13: Revise is the same as response 12.
Point 14: - Lines 244-246) "However, the FBG has structure that supports the activity of hip extensors, knee extensors, and ankle dorsiflexors, which are antagonistic to these muscles and the activity of hip extensors, knee extensors, and ankle dorsiflexors during the early stance." This sentence has no sense.
Response 14: I agree with your comment. I deleted the sentence.
Lines 244-246:
However, the FBG has structure that supports the activity of hip extensors, knee ex-tensors, and ankle dorsiflexors, which are antagonistic to these muscles and the activity of hip extensors, knee extensors, and ankle dorsiflexors during the early stance.
Point 15: - Lines 252 -253) "In addition, preventing a decline in walking speed can be expected to reduce the risk of adverse events and enjoyment the beneficial effects of walking."
Remove in addition: "Preventing a decline in walking speed can be expected to reduce the risk of adverse events and ENJOY the beneficial effects of walking."
Response 15: I deleted the sentence.
Lines 473:
In addition, pPreventing a decline in walking speed can be expected to…
CONCLUSIONS:
Point 16: - I think that here you can extend a bit more the sentence "The FBG may be useful as a tool to promote health activities." and thus emphasize the relevance of application of the FBG.
Moreover, in this section you can elaborate on the importance of this work or suggest extensions.
Response 16:
Revised line 494:
Tthe FBG may be useful as a tool to promote health activities to support the walking.
Added sentence Lines 491-494:
For the above reasons, our FBG has a biomechanical effect. The FBG may be able to maintain step length and prevent a decline in walking speed and enjoyment the bene-ficial effects of the walking in healthy persons. Therefore, the FBG may be useful as a tool to support the walking.
Bibliography:
Point 17: - 3. author name: uppercase letter. remove the first name to be consistent with the other articles.
- article title is in uppercase
Response 17:
Revised references and reference number
- Nemoto K. Effects of High-Intensity Interval Walking Training on Physical Fitness and Blood Pressure in Middle-Aged and Older People: Commentary. Mayo Clin Proc 2007, 82, 803–811, doi:10.1097/01.ogx.0000312157.35839.ce.
- Meads, C.; Exley, J. A systematic review of group walking in physically healthy people to promote physical activity. J. Technol. Assess. Health Care 2018, 34, 27–37, doi:10.1017/S0266462317001088.

Round 2
Reviewer 1 Report
The Aurors made quite a few adjustments that helped improve the level of the study. The corrections made in the introduction, methodological procedures are appropriate. I recommend adding a legend / comment to all tables that are very large and confusing. The conclusion is a correction, but it still seems to me not entirely specific, so that it is a direct link to the results. I recommend making further improvements, then re-proofreading and the article could then be published.
Author Response
Response to Reviewer 1 Comments
Dear Dr. Reviewer 1
Thank you very much for reviewing our manuscript and offering valuable advice so many times.
Point 1: I recommend adding a legend / comment to all tables that are very large and confusing.
Response 1:
Make it easy to see, I devided table3 and table4.
I added table 2-7 title and legend.
Line 158-167: I revised the sentence.
Table 2. Pre and post wearing of FBG, average (SD), paired t-test (p< 0.05).
Table 3. Effect of wearing comparison FBG and CG, average (SD), two-way ANOVA (p<.05).
Table 4. Pre and post wearing of FBG:DJS, average (SD), paired t-test (p< 0.05).
Table 5. Effect of wearing comparison FBG and CG DJS average (SD), two-way ANOVA (p<0.05).
Table 2. Spatio-temporal data during walking pre and post wearing of FBG. average (SD), paired t-test, p values < 0.05 are bolded.
Table 3. Kinematics Kinetics data, and Ground reaction force during walking pre and post wearing of FBG. average (SD), paired t-test, p values < 0.05 are bolded.
Table 4. Spatio-temporal data, during walking under two garments condition, i.e., effect of wearing comparison FBG and CG, average (SD), two-way ANOVA, p values < 0.05 are bolded.
Table 5. Kinematics, Kinetics data and Ground reaction force during walking under two garments condition, i.e., effect of wearing comparison FBG and CG, average (SD), two-way ANOVA, p values < 0.05 are bolded.
Table 6. Dynamic joint stiffness for pre and post wearing of FBG. Post wearing of the FBG had significantly decrease dynamic joint stiffness values in the Terminal stance when compared to the pre wearing condition. average (SD), paired t-test (p< 0.05). p values < 0.05 are bolded.
Table 7. Dynamic joint stiffness during walking under two garments condition, i.e., effect of wearing comparison FBG and CG. The FBG group had significantly lower Hip Dynamic joint stiffness values in the Terminal stance than the CG groups. average (SD), two-way ANOVA, p values < 0.05 are bolded.
Point 2: The conclusion is a correction, but it still seems to me not entirely specific, so that it is a direct link to the results.
I deleted and revised the conclusion.
Conclusion:
The FBG decreased hip DJS in the terminal stance and affected walking speed. Wearing the FBG increase walking speed, and the decrease in the peak knee joint flexion angle during the initial-swing and the decrease in the peak knee extension moment during the loading-response. In addition, the FBG decreased hip DJS in the terminal stance. The passive elastic moment by the high elasticity part of the hip joint front in the FBG supported the internal hip flexion moment. For the above reasons, our FBG has a biomechanical effect. The FBG may be able to maintain step length and prevent a decline in walking speed and enjoyment the beneficial effects of the walking in healthy persons. Therefore, the FBG may be useful as a tool to support the walking.
